# Disaggregation Model: A Novel Methodology to Estimate Customers' Profiles in a Low-Voltage Distribution Grid Equipped with Smart Meters

Guilherme Ramos Milis [1,2,*], Christophe Gay [2], Marie-Cécile Alvarez-Herault [1] and Raphaël Caire [1,*]

1    Grenoble Electrical Engineering Laboratory (G2Elab), CNRS, University Grenoble Alpes, 38000 Grenoble, France; marie-cecile.alvarez@g2elab.grenoble-inp.fr
2    Enedis, 92400 Courbevoie, France; christophe.gay@enedis.fr
*    Correspondence: guilherme.ramos-milis@grenoble-inp.fr (G.R.M.); raphael.caire@grenoble-inp.fr (R.C.)

**Abstract:** In the context of increasingly necessary energy transition, the precise modeling of profiles for low-voltage (LV) network consumers is crucial to enhance hosting capacity. Typically, load curves for these consumers are estimated through measurement campaigns conducted by Distribution System Operators (DSOs) for a representative subset of customers or through the aggregation of load curves from household appliances within a residence. With the instrumentation of smart meters becoming more common, a new approach to modeling profiles for residential customers is proposed to make the most of the measurements from these meters. The disaggregation model estimates the load profile of customers on a low-voltage network by disaggregating the load curve measured at the secondary substation level. By utilizing only the maximum power measured by Linky smart meters, along with the load curve of the secondary substation, this model can estimate the daily profile of customers. For 48 secondary substations in our dataset, the model obtained an average symmetric mean average percentage error (SMAPE) error of 4.91% in reconstructing the load curve of the secondary substation from the curves disaggregated by the model. This methodology can allow for an estimation of the daily consumption behaviors of the low-voltage customers. In this way, we can safely envision solutions that enhance the grid hosting capacity.

**Keywords:** load models; low-voltage grid; load curve; disaggregation model; optimization; curve fitting; K-means; PCA



## 1. Introduction

The international negotiations on climate policies reveal that we are grappling with the impacts we have on the environment. One way to mitigate these impacts directly involves reducing the use of fossil fuels and increasing electrification. This leads to changes in the level that the distribution electrical network is used. These changes extend from the generation and distribution of energy to the way customers use it. They are part of a phenomenon called energy transition. Its effects can be immediately seen when we look at the new directive of the European Union, "Fit for 55", which aims to reduce greenhouse gas emissions by 55% by 2030. To achieve this goal, one of the proposals is to increase the renewable energy target from 32% to at least 40% by 2030 [1].

We can also take a closer look at the energy transition by examining the French scenario in more detail. According to data from the DSO Enedis, responsible for energy distribution in 95% of metropolitan France, the number of production sites connected to the grid increased by 178% between the years 2012 and 2023. The same data show us that the total number of electric vehicle charging infrastructures connected to the grid grew by 2000% between 2015 and 2023 [2]. The French Ministry of Ecological Transition estimates the distribution of residential energy use in France by type. The heating system represents approximately 70% of the country's residential energy consumption. Among

the various energy sources used for residential heating, electric sources have been growing proportionally year after year, as indicated by these data [3]. We can therefore observe that the fight against environmental impacts is reflected by a significant change in the habits of electricity production and consumption, centered in the distribution grid.

In this scenario of the strong integration of DERs into the distribution network, being able to better estimate the hosting capacity of the network becomes vital for DSOs to better guide and manage network investments. This topic has been explored in numerous publications that investigate methodologies for assessing the network's hosting capacity [4–6]. Looking at them in detail, we can observe that consumer load models of the network are one of the key elements impacting the hosting capacity assessment. Therefore, we can understand that the development of more accurate models using smart meter data can provide us with a better estimate of this factor, allowing us to better integrate DERs into the distribution network.

As the uses of the grid evolve, the distribution grid also evolves to keep pace with these changes. The electrical grids are becoming more digital, transforming our networks into smart grids. According to the International Energy Agency, "smart grids are electrical networks that use digital technologies, sensors, and software to better match the supply and demand for electricity in real-time while minimizing costs and maintaining the stability and reliability of the network". The major enabler of this digitization lies in the replacement of electricity meters, transitioning from analog/digital meters to smart meters. Smart meters are electronic devices that record information, such as electrical energy consumption, and communicate this information to the consumer, electricity providers, and DSOs.

We can look at some examples of this evolution in the metering landscape around the world. The Italian DSO ENEL initiated the deployment of its first generation of smart meters in 2001 for your 32 million customers [7]. In 2016, the Italian regulatory body defined the minimum functions for the new generation of meters. This new generation of meters began to be distributed in 2017, with the aim of making energy consumption more sustainable and becoming a key element in the construction of smarter and more circular cities [8]. The U.S. DSO PG&E launched the SmartMeter™ program for the installation of smart meters in 2006, and the majority of its customers were equipped with them by 2012 [9]. The Canadian DSO BC Hydro is the primary electricity distributor in the British Columbia region, serving over 4 million customers in most areas of the province. The deployment of smart meters began in 2011 and was completed in 2012 [10]. The Korean Electric Power Corporation, KEPCO, is one of the largest energy companies in South Korea. Since 2012, the KEPCO has installed approximately 120,000 smart meters to promote the smart grid infrastructure. Until 2018, 6.8 million households in South Korea were equipped with smart meters. The goal was to reach 22.5 million households by the early 2020s [11]. TEPCO, the largest Japanese DSO, completed the installation of approximately 29 million smart meters by the end of 2020 [12]. The French DSO Enedis began rolling out its meters in 2015 and, from 2023 last figures, has over 37 million smart meters deployed [13]. It is possible to observe that the change in the metering landscape is a global phenomenon, occurring in parallel with changes in grid usage.

The exploitation of smart meter data makes it possible to explore new solutions that helps the grid to receive this new usage in the energy transition context. However, the exploration of this data requires some attention and care. In the European Union, consumer load profile data from smart meters are considered personal data by the General Data Protection Regulation. In this way, the use of this personal data requires the explicit consent of each consumer for its use. Furthermore, even with consent for use, these data have a duration limit for which they can be retained as stipulated by the GDPR. Another possible impediment is related to the fact that classic smart metering infrastructures have limitations to obtain all the low-voltage consumers' load profiles due to the communication protocols employed, such as the power carrier line G3 used in France. Challenges related to Big Data in smart grids are increasingly being discussed, such as the data size that can reach petabytes, as well as the costs and environmental impacts of developing data

centers capable of storing this volume of data [14–16]. Therefore, the development of new profiles in a scenario of evolving network usage becomes extremely important for DSOs. The objective of this paper is to propose a novel methodology to estimate the load curves of consumers on the low-voltage network using very few data from smart meters. This modeling will be based on data provided by the French DSO Enedis, but the model had been built to be compliant with data from most smart metering infrastructure. The possible constraints that may be faced when using smart meter data for load curve estimation are related to data privacy concerns and IT infrastructure limitations. The aim of this model is to explore the use of smart meter data for profile or load curve estimation and be a model that can be employed to assist in the integration of new usages. To achieve this, the model starts with the load curve of the secondary substation and disaggregates it into individual load curves of customers, as illustrated in Figure 1. The idea behind this model is to perform daily disaggregation, allowing us to access the customer-level load curve every day. The customer load curve is the input data for network calculation tools, enabling us to compute voltage and power transits in the network. With this daily customer-generated load curve data, it is possible to better estimate the hosting capacity of the network, as well as the available flexibility of the demand.

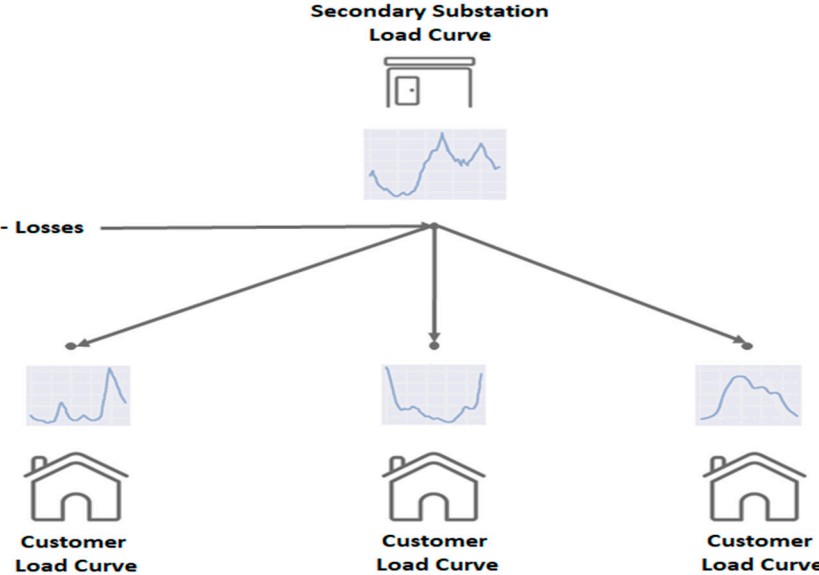

**Figure 1.** Example of obtaining profiles at the customer level from the disaggregation model. Where "secondary substation" refers to a MV/LV transformer, while the "customer" refers to a LV consumer.

Looking at the literature, T. Barbier [17] conducted a classification of different types of electrical consumption models. This classification is based on two factors. The first factor is a framework with three levels of consumption, including the individual level of the customer and their devices, the aggregated level by type of customer and type of device, and the aggregated level of the customer by spatial zone and their devices. The second factor is a framework of variables influencing consumption, including the device inventory, the technology of these devices, usage behavior, socioeconomic level, meteorology, variables related to the place of consumption, customer subscription, and the electricity market. Thus, he lands at six categories of models.

The first category includes models based on aggregated consumptions in a zone. This category includes models where the scale is the level of electrical consumption aggregated by spatial zone. These models primarily focus on the temporal component of consumption and are commonly referred to as "forecasting" models. In this category, refs. [18–21] stand out in the analysis and evaluation of this type of forecasting model.

The second category consists of models based on consumptions by the type of customer. This category includes models where the scale is the level of consumption per type of

customer. This type of modeling produces different load curves for each type of customer (residential, industrial, etc.), and customers of the same type will have identical profiles, adjusted in energy [22–24]. This category can be subdivided based on the model's input data.

According to the author [17], the third category consists of models based on consumption by usage. This implies that these models based on usage types can be significantly influenced by external factors other than consumers, such as temperature. This is illustrated by an example of such a model that decomposes thermo-sensitive uses from non-thermosensitive uses based on load curves from different zones [25]. It illustrates the example of thermosensitivity in France, allowing for the Transmission System Operator and Distribution System Operators to understand the influence of meteorological hazards on network sizing. The work presented in [26] decomposes individual consumption loads based on their thermal sensitivity. This allows for an estimation of the portion of the load that comes from thermosensitive uses and the portion that comes from non-thermosensitive uses. An analysis is conducted in [27] to correlate load profiles measured through smart meters with local temperature variations to extract heating/cooling patterns as a function of the outdoor temperature. In addition to temperature, we can cite external factors of global magnitude, such as the COVID-19 pandemic. It has changed people's daily lives due to social distancing measures, which affected the way people consumed electricity. Studies on how this external factor influenced consumption patterns are depicted in [28–30].

The next category consists of models based on consumption by the type of customer and usage. This category essentially includes models for the residential sector that are based on consumption by the type of use of equipment in the sector. This category comprises models [31–33] that involve disaggregating electrical consumption profiles with information on the ownership rate of devices and possibly with socio-economic data. In it, we can also find residential models based on the technological advancements of equipment and new types of usage. In [34], a study of emerging technologies in residential load profiles is proposed. The work conducted in [35] analyzes the impacts of low-carbon technologies, such as electric vehicles and photovoltaics, on the network.

The fifth category consists of models based on consumption per customer. According to the author [17], models belonging to this category use data from smart meters to make short-term predictions of electrical consumption for individual customers [36–38].

The last category consists of models based on consumption per device. This category aims to estimate consumption at different levels through the aggregation of device consumption. An example of this is Non-Intrusive Load Monitoring models [39–41]. The idea is to have a sufficiently fine temporal resolution of measurements to capture the energy signatures of different devices.

Based on this analysis of the classification of different types of electricity consumption models conducted in [17], we can conclude that "bottom-up" and "top-down" models are the ones that come closest to the initial idea of disaggregation models. Focusing in these types of residential consumption models, in [21], the author aims to provide a review of various modeling techniques employed to construct residential consumption models. He then divides these models into two categories, top-down and bottom-up models. Top-down models are those that use the estimation of residential sector consumption and other attributes to produce estimates of consumption at the individual level of residences. On the other hand, bottom-up models start from the estimation of individual or group consumption to arrive at consumption in larger networks, such as national or regional. In [21], top-down models are subdivided into econometric models and technological models. Meanwhile, bottom-up models are subdivided into statistical models and engineering models. The review [31] proposes an analysis of residential load curve models. The article complements the characterization of bottom-up and top-down models, where one uses microscopic data as input for the residential level and the other uses macroscopic level data as input. In [31], the division of models is made into top-down models, bottom-up models, and hybrid models. Knowing that one-third of final electricity consumption is

residential in Europe, along with the evolution of network uses through new equipment and the electrification of others, publication [42] presents a review of residential load profile models. The article analyzes the models in four different ways. The first is by the type of method used, the second is about the sampling rate of the profiles, the third is about the type of application proposed by the model, and the fourth is about the statistical techniques employed. Looking more closely at the methods used, we observe that the author in [42] subdivides them into top-down, bottom-up, and hybrid approaches.

We can understand that our disaggregation methodology does not precisely fit into these categories of residential models studied in the literature. The scale at which we intend to build the disaggregation model is that of the secondary substation level to assess the load curve or profile of each LV customer attached to it. This is not clearly represented in the literature. The load curves of customers are the inputs for the DSO's tools to carry out network planning studies. Therefore, estimating them accurately becomes a necessary task in this context of evolving network usage. The use of usage decomposition methodologies, such as "bottom-up", proves to be extremely dependent on measurements with a fine temporal resolution. DSOs may not necessarily have this degree of precision in their measurements. Our disaggregation model avoids this situation, making it more accessible.

This literature review confirms that we are seeking to undertake innovative research that will contribute to expanding the horizons of the field. Therefore, the work presented in this publication aims to contribute to the construction of a disaggregation model based on data from the secondary substation and maximum power consumption customers' data. This methodology, based on real data from the low-voltage distribution network, can lead us to a more realistic representation of the different consumption profiles of network consumers and their related technical constraints on the electrical infrastructure. These low-voltage consumption profiles are swiftly evolving as new technologies appear, such as electric vehicles, residential photovoltaics, and storage units. This makes the use of historical profiles constructed by DSOs outdated. The proposed methodology, based on data modeling, proposes a mathematical and dynamic approach that supports the emergence of congestion or over voltages in LV networks. With profiles closer to the reality of low-voltage consumption, we can make a better estimation of the network's hosting capacity, thus helping to best integrate new loads into the network or support smart network reinforcement together with flexibility support from Distributed Energy Resources.

## 2. Materials and Methods

The general principle of the model is to disaggregate the load curve of the substation into load curves for each downstream customer. To achieve this, we will carry out this process in two steps:

- Customer Segmentation: The N customers of a substation share similar characteristics, either because they belong to the same categories (residential, professional, etc.) or because they have similar consumption habits. From this perspective, the segmentation (or the number of clusters) of these customers is performed to identify K groups of similar customers among the N customers of the substation, where K < N.
- Secondary Substation Load Disaggregation: The load curve of the substation is then disaggregated into K curves, representing the K groups of similar customers at the substation. These K curves are then adjusted in energy to assign to each customer the curve of the group to which they belong.

The data scope chosen for the development of this model included 48 secondary substations, where load curve data were collected for the period from March 2022 to March 2023. Additionally, maximum power and energy data from Linky smart meters (a total of 5318 m) were collected downstream of these substations for each day of the period. It is essential to specify that the Linky data of the customers used in this study are associated with a customer panel, who have given their consent to Enedis for the use of their data in the context of network studies, including the one integrated into this article, in accordance with

the General Data Protection Regulation (GDPR). Therefore, these data are not accessible beyond this perimeter. We have three types of information per secondary substation:

- The load curve of the substation.
- The maximum power value in watts measured by Linky for all customers connected in the substation.
- The time of day (hours and minutes) when the maximum power occurred.

With all this information in mind, we can delve into the details of each step of the model.

## 2.1. Customer Segmentation

The objective of this step is to identify *K* groups of similar customers at a substation with N customers. The idea behind this is to simplify the subsequent disaggregation step, as we will disaggregate the substation's load curve into fewer curves (*K* < N). Additionally, working at an individual customer level is extremely complex due to the significant variability in consumption. Therefore, working at a slightly higher level (*K* groups of similar customers) is considered the optimal strategy. Various temporal scales could be used to segment customers at a substation, such as daily, by days of the week, monthly, seasonal, or yearly, among others.

Segmentation by season was chosen as it offers more advantages. It provides a sufficient volume of data for customer segmentation and allows for visualization of the evolution of the behavior of all customers over time. Customers who exhibit similar behavior within a given season may not necessarily show similar behavior in other seasons.

The data selected for building this dataset are maximum power data. They include the exact maximum power and related time stamp of it. The decision to use only these data is to simplify the construction process by leveraging one of the pieces of information collected by the Smart Metering Infrastructure. Moreover, the simpler the information used in model construction, the easier it will be to reproduce it in other research. The maximum power data enable construction of the dataset used in this segmentation step. This dataset is represented in Table 1.

**Table 1.** Dataset constructed for the segmentation step.

| Customer | Maximum Power Day in 1 | Occurrence in Day 1 | Maximum Power Day in 2 | Occurrence in Day 2 | Maximum Power Day in D | Occurrence in Day D |
|---|---|---|---|---|---|---|
| 1 | Value (kW) | (hh:mm) | Value (kW) | (hh:mm) | Value (kW) | (hh:mm) |
| 2 | Value (kW) | (hh:mm) | Value (kW) | (hh:mm) | Value (kW) | (hh:mm) |
| ... | ... | ... | ... | ... | ... | ... |
| N | Value (kW) | (hh:mm) | Value (kW) | (hh:mm) | Value (kW) | (hh:mm) |

We can observe that for the N customers connected to a secondary substation, the maximum power values and their occurrence time are represented for all D days of the respective season.

To preprocess the data, the dimensionality reduction technique called Principal Component Analysis (PCA) was employed [43]. PCA is a statistical technique used to reduce the dimensionality of data while preserving, as much as possible, the essential information contained in the data. Specifically, this method is beneficial for reducing the amount of processed data, limiting data redundancy, and facilitating easier data visualization, leading to a better understanding [43].

In the literature, various methods exist for performing segmentation. For the implementation of this disaggregation model step, we chose to work with partition methods, specifically K-means [44], due to their simplicity in understanding the proposed results. K-means clustering is one of the most well-known and widely used unsupervised learning algorithms. Generally, unsupervised algorithms make inferences from datasets using only

input vectors without reference to known or labeled outcomes [45]. The goal of K-means clustering is to group similar data points and discover underlying clusters. To determine the number of groups K that will be assigned in our dataset, the elbow method [46] is employed. It aims to identify the number of clusters K that strikes a good balance between reducing the sum of squared errors (SSE) and the complexity of the model. It is an easy-to-understand and easy-to-apply method, making it a practical tool for choosing K. However, the drawback is the subjectivity of the method, as identifying the elbow can be challenging. For this reason, we chose to use the Python library Kneed [47] to automatically detect elbows and eliminate the subjective aspect. Figure 2 illustrates the step of customer segmentation.

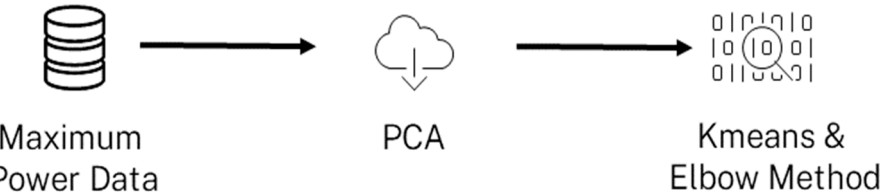

**Figure 2.** Customer segmentation steps.

### 2.2. Secondary Substation Load Disaggregation

The objective of this step is to formulate a method to disaggregate the load curve of a secondary substation into *K* curves that represent the *K* groups among the N customers identified in the segmentation step. To begin thinking about how to achieve this solution, let us start by formulating the problem mathematically.

$$Load\ Curve_{\substack{secondary \\ substation}} = \sum_{k=1}^{K} Cluster\ Curve\ (k) \tag{1}$$

As shown in Equation (1), the disaggregation problem implies that the sum of *K* curves (representing the groups of customers found in the segmentation process) is equal to the curve of the secondary substation. Approaching the problem from this perspective, once the equations describing the *K* curves are defined, this problem can be solved by applying a curve-fitting method. Curve-fitting is a commonly used technique in data analysis to estimate the parameters of a mathematical function that best describe a set of data [48]. Its basic principle is to find the optimal parameters of a mathematical function that minimizes the square of the difference between the observed data and the values predicted by the function, as shown in Equation (2):

$$min\left( \sum_{i=1}^{M} (y_i - f(x_i, \theta))^2 \right) \tag{2}$$

The goal is to find $\theta_{optimal}$ values that minimize quadratic error, called the cost function. In other words, curve fitting involves solving an optimization problem where the objective is to identify the values of the parameters $\theta$ that make the cost function as small as possible. To solve this curve-fitting problem, the Python library lmfit (version 1.2.2) [49] was chosen. As explained by the developers, lmfit provides a user-friendly interface for defining models, specifying parameters, fitting data, and retrieving results. It also allows for the choice of a variety of optimization methods. In the case of this disaggregation problem, the chosen method is L-BFGS-B (Limited-memory Broyden–Fletcher–Goldfarb–Shanno with Bound constraints) [50]. With the definition of how the disaggregation problem will be approached, it is now necessary to define which mathematical function best describes our data. The data on which the curve-fitting will be performed are the daily load curves of the secondary substations.

### 2.2.1. Function One—Pure Mathematical Model

The option chosen to address the question of which equations best describe the sum of *K* curves in Equation (2) is a sum of N Gaussians, as shown in Equation (3).

$$f(x, a, b, c) = \sum_{k=1}^{K} \left( a_k * exp^{\frac{-(x-b_k)^2}{2*c_k^2}} \right) \tag{3}$$

where formally θ in the function f(x, θ) now corresponds to the parameters $a_k$, $b_k$, and $c_k$. $a_k$ determines the height of the Gaussian, $b_k$ determines the position of the center of the Gaussian, and $c_k$ determines the width of the Gaussian.

The reason that leads us to believe that this assumption is a good starting point is as follows: if we examine the shape of customer load profiles [51,52], the consumption peaks at different times of the day resemble Gaussian behavior. These profiles represent the behavior of a group of customers belonging to the same class. Furthermore, a Gaussian is a natural candidate because it is a classic function in statistics related to the central limit theorem. This theorem plays a fundamental role in statistical theory by showing that the mean of many independent and identically distributed random variables approximately follows a Gaussian distribution, regardless of the initial distribution of these variables. This demonstrates its ubiquity in modeling random phenomena, such as the electrical consumption of low-voltage customers.

### 2.2.2. Function Two—Adding Electrical Properties to the Purely Mathematical Model

The idea behind this function is to use the information available thanks to Linky meters to approximate the modeling of the real behavior of network customers. In order to do this, we begin by analyzing the maximum power occurrence (see Table 1) data measured by the Linky meters. If we look at this information for each individual customer throughout an entire season, we can attempt to extract trends about these occurrences that can guide us in the formulation of Function 2. Therefore, the occurrences of maximum power for five customers during the days in the winter season are presented in Figure 3.

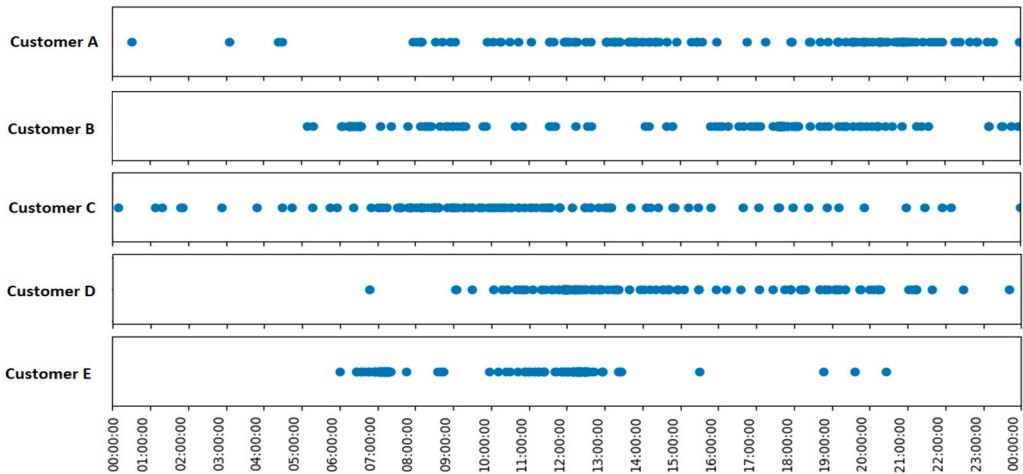

**Figure 3.** Occurrences of maximum power for five customers during the days in the winter season.

It can be observed in Figure 3 that each customer exhibits different consumption patterns, and they vary considerably throughout the season. This leads us to believe that observing the occurrences of maximum power for a group of multiple customers could reveal patterns that are easier to identify. In order to confirm this, all occurrences of maximum power (time of day when the maximum power value was recorded by Linky infrastructure) for two groups of customers are observed. Group 1 consists of 70 customers, and group 2 has 25 customers. This observation is presented in Figure 4 through a histogram, where each bar represents the total number of maximum power

occurrences, of all customers in the group during the winter season, which occurred within a specific time interval of the day. The intervals have a duration of 10 min, resulting in a total of 144 intervals.

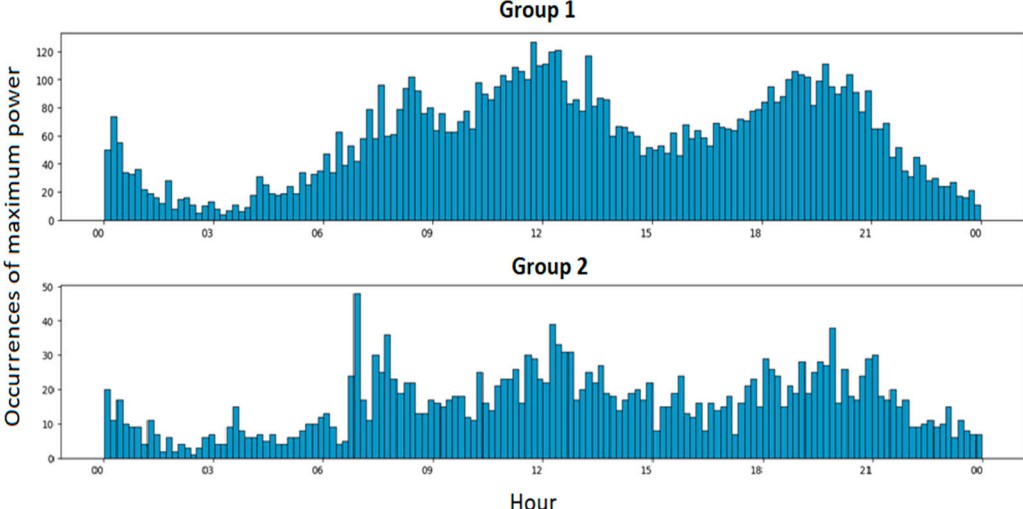

**Figure 4.** Occurrences of maximum power for all customers of each group during the winter season.

Observing the occurrences by group makes the trends more apparent. For group 1, four concentrations of maximum power occurrences can be noted: around noon, 8 pm, 8 am, and midnight. For group 2, there is a significant concentration at 7 am and less pronounced concentrations around noon and 8 am. As the previous stage of disaggregation segments the customers from the secondary substation, we will have formed the groups of customers to analyze. Now, it is necessary to translate this information into an adjustment of the parameters of our function.

The concentrations of maximum power at a given period indicate whether it is a significant time for the cluster, deserving high resolution to finely adjust the peaks. Therefore, for each cluster, we can say that the most pronounced concentrations in the histograms will indicate the maximum number of Gaussians, as well as the initial value of the "b" parameter for each of these Gaussians. To identify these concentrations, the "find_peaks" function from the Python SciPy library was used [53]. It allows for the identification and localization of local maxima in a one-dimensional dataset through a simple comparison of neighboring values. The points of maximum concentration of the histogram will be denoted by the letter g. The time of the day at which these points appears will be denoted as $h_g$. This guides us towards another way of expressing our sum of the Gaussian function f(x, a, b, c) and adjusting the parameter bounds. For each cluster k, we will have G numbers of Gaussians found from the histogram of maximum power occurrences, as indicated by Equation (4). Since the number G of Gaussians can be different for each cluster k, it will be denoted as $G_k$.

$$f(x, a, b, c) = \sum_{k=1}^{K} \sum_{g=1}^{G_k} \left( a_{t,k} * exp^{\frac{-(x-b_{t,k})^2}{2*c_{t,k}^2}} \right)$$

$$0 \leq a_{g,k} < max(Secondary\ Substation\ Curve)$$
$$h_g - 0.5\ hours \leq b_{g,k} \leq h_g + 0.5\ hours$$
$$1\ hour < c_{g,k} < 6\ hours$$

(4)

Having the parameter $a_{t,k}$ presenting a possible value of zero allows for the optimization to discard the Gaussian (t,k) if it is not significant for curve-fitting, setting its height to zero. This accounts for the variable behavior indicating that customers in this group may not have their maximum power at that time every day. The parameter $b_{t,k}$ capable of varying by about half an hour around the time $h_t$ allows for the center of the Gaussian to

be moved, which aligns with the occurrences of maximum power around each maximum. The parameter $c_{t,k}$ represents the width of the Gaussian, with the lower bound set to one hour. This prevents the presence of peaks that appear and disappear very quickly, as we assume that the consumption of the customer group has some inertia in its variations. The upper bound for the $c_{t,k}$ parameter is limited to 6 h to avoid losing information about a possible important peak.

### 2.3. Global Vision of the Model

To facilitate the overall view of the model, a schematic gathering all of its stages is presented in Figure 5.

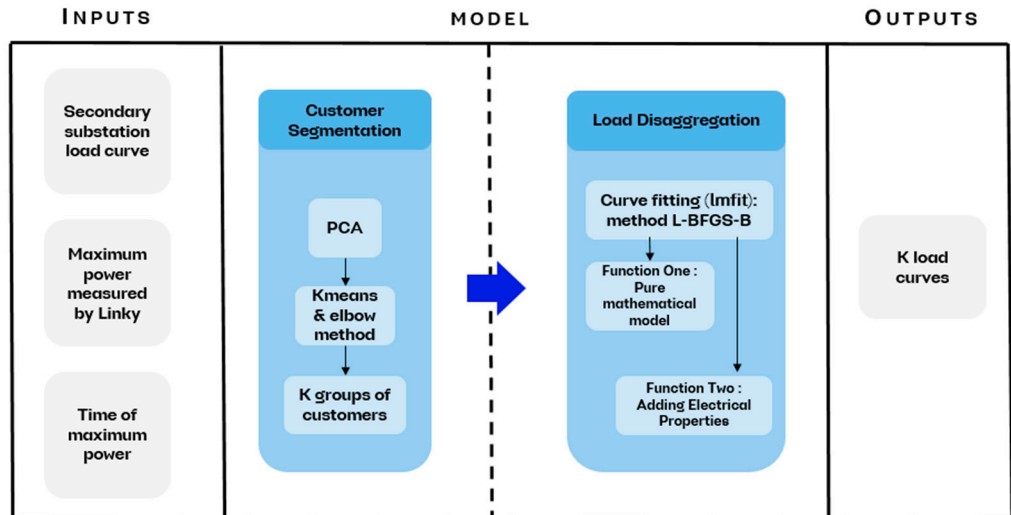

**Figure 5.** Schematic of the disaggregation model.

### 2.4. Error Evaluation

The error produced by the model is calculated using the symmetric mean average percentage error (SMAPE). SMAPE is a commonly used metric to assess the accuracy of forecasts or prediction models. It calculates the accuracy of predictions by comparing actual and predicted values in a symmetric way, as shown in Equation (5):

$$SMAPE = \frac{1}{N} \times \sum_{n=1}^{N} \frac{|P_n - R_n|}{(|R_n| + |P_n|)/2} \tag{5}$$

where P is the predicted value and R is the real value. Therefore, for each data point, it takes the absolute difference between the actual value and the predicted value and then divides it by the sum of the actual and predicted values. SMAPE is robust to small values. Due to its symmetry, SMAPE ensures that the penalty for error is not sensitive to large differences between the prediction and the actual value.

With the help of SMAPE, we will evaluate the error between the reconstruction of the load curve of the secondary substation based on the model's result and the true load curve of the secondary substation. Additionally, expressing the error as a percentage makes the differences between the results obtained from the functions presented in Section 2.2 more understandable to the reader. In this way, we will be able to assess how well the disaggregation model can reconstruct the load curves of the secondary substation.

## 3. Results

The results of the disaggregation model are presented for both functions described in Section 2.2. The evaluation of the results for both models is conducted along two cases. In the first case, a random secondary substation is selected, and the model is applied to the

load curve for the winter season. In the second case, the error is computed for all secondary substations on all days of the winter season.

### 3.1. Results for One Random Secondary Substation

A secondary substation was randomly chosen from the set of 48 in our dataset. This secondary substation has 198 connected consumers. The result of consumer segmentation, presented in Section 2.1, led us to the number *K* of similar customer groups equal to 6. The maximum power data from these 198 customers are used to construct the input dataset for our segmentation process, as presented in Section 2.1. By applying the segmentation steps as described, we obtain a number *K* of similar customer groups equal to 6. This means that these 198 customers can be segmented into six groups, where each group consists of customers with similar maximum power values and similar occurrence times of these powers throughout the winter season. The quantity of customers in each group can be observed in Table 2.

**Table 2.** Number of customers per group after the segmentation step.

| Group | 0 | 1 | 2 | 3 | 4 | 5 |
|---|---|---|---|---|---|---|
| Customers | 72 | 74 | 5 | 3 | 24 | 20 |

For each day of the winter season, our disaggregation model is put into practice using the two functions described in Section 2.2 and the six groups of customers. The results of the errors calculated are presented in Table 3.

**Table 3.** SMAPE errors of the curve-fitting for the winter season.

| SMAPE | Function 1 | Function 2 |
|---|---|---|
| Minimal (%) | 4.09 | 1.60 |
| Mean (%) | 6.36 | 2.64 |
| Maximum (%) | 10.81 | 3.99 |

In order to look at the load curves produced by the model with both functions, the days with minimal SMAPE error are selected. These curves are illustrated by Figure 6.

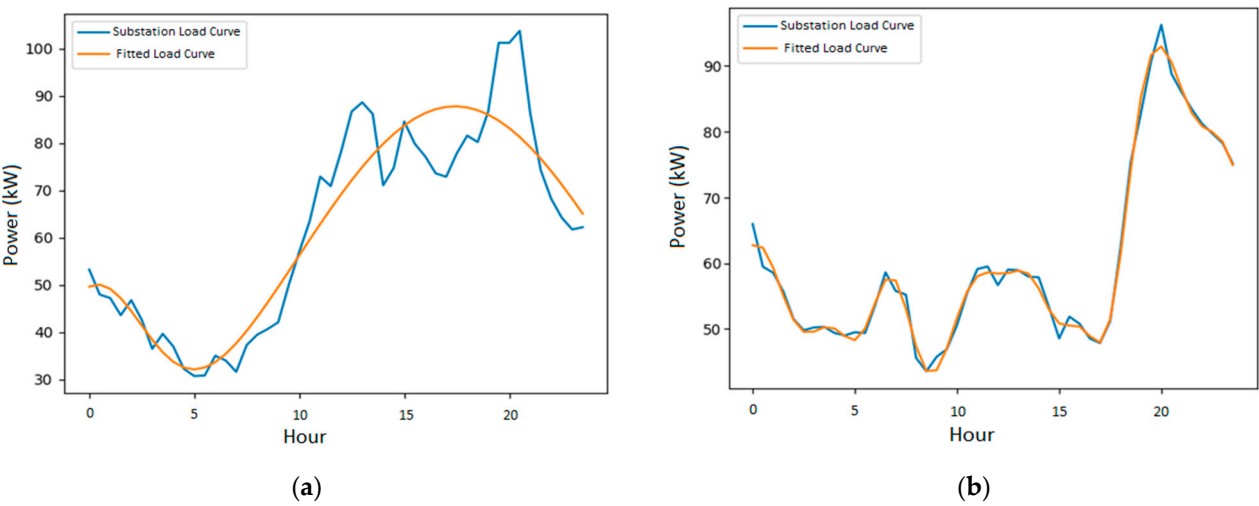

(**a**)                                                       (**b**)

**Figure 6.** (**a**) Minimum SMAPE result produced by the model using function one; (**b**) minimum SMAPE result produced by the model using function two.

The individual load profiles of each cluster are presented below. The individual load profiles produced by the model using function one and two are presented by Figures 7 and 8, respectively.

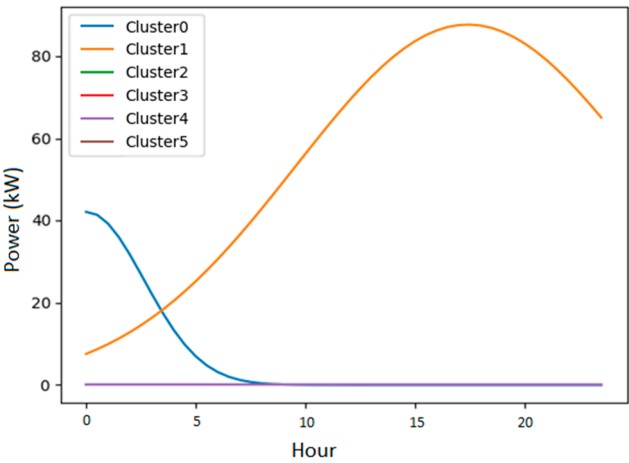

**Figure 7.** The individual load profiles of each cluster produced by the model using function one.

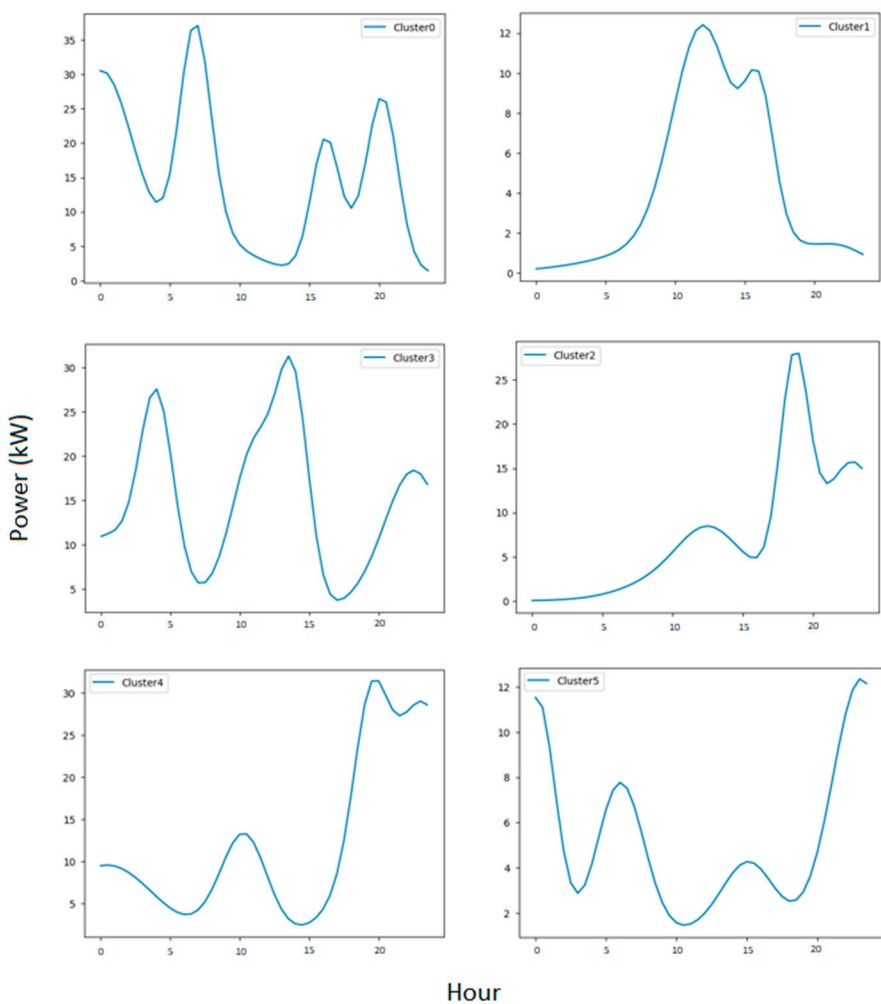

**Figure 8.** The individual load profiles of each cluster produced by the model using function two.

### 3.2. Results for All Secondary Substations in the Dataset

For the 48 secondary substations, the total quantity of groups obtained in the segmentation step is presented in Table 4.

**Table 4.** Number of secondary substations per quantity of groups after the segmentation step.

| Quantity of Groups | 4 | 5 | 6 | 7 |
|---|---|---|---|---|
| Number of Substations | 4 | 23 | 16 | 5 |

For each day of the winter season, our disaggregation model is put into practice using the two functions described in Section 2.2. The results of the errors calculated using SMAPE are presented in Table 5.

**Table 5.** Mean SMAPE errors of the curve-fitting for all secondary substations in the winter season.

| SMAPE | Function 1 | Function 2 |
|---|---|---|
| Minimal (%) | 8.43 | 2.93 |
| Mean (%) | 17.86 | 4.91 |
| Maximum (%) | 60.15 | 7.08 |

In order to look at the load curves produced by the model with both functions, the days for the secondary substations with minimal SMAPE error are selected. These curves are illustrated by the following Figure 9.

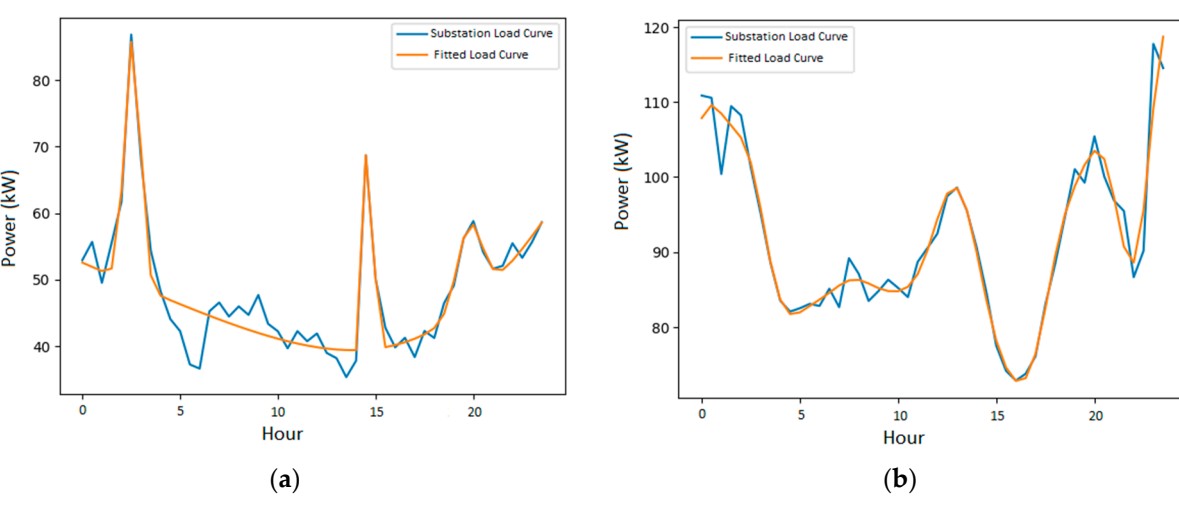

(**a**)          (**b**)

**Figure 9.** (**a**) Minimum SMAPE result produced by the model using function one; (**b**) minimum SMAPE result produced by the model using function two.

## 4. Discussion

The results show us that the disaggregation model using function 2 presents better outcomes, even though function 2 has fewer degrees of freedom than function 1. If we look closely at the results with both functions for one random secondary substation, we can observe that function 1, purely mathematical, manages to capture the large peaks and valleys of the load curve, as shown in Figure 6a. However, it fails to capture the secondary peaks, thus leaving the curve-fitting result unsatisfactory. On the other hand, function 2 incorporates characteristics of customer consumption into the modeling, thanks to the data from Linky meters. The interest in considering the behavior of customers from the segmentation step is to ensure consistency with reality, thus bringing the model closer to what happens in the network. It can be observed in Figure 6b that the fitted curve has successfully captured the various consumption peaks represented by the secondary substation load curve. This result is reflected in the shapes of individual profiles, as shown in Figure 8. This highlights the consistency between the distribution of maximum power occurrences per group of customers and the consumption measurements of all customers observed based on the secondary substation load curve. This is reflected when comparing

the errors of the functions. The error produced by function 2 is lower than that of function 1 as shown in Table 3.

The result obtained for a random secondary substation holds when we look at all the substations in our dataset. As presented in Table 5, function 2 produces results with lower errors than function 1. This assures us of the importance of the maximum power information measured by the Linky meter. Additionally, it underscores the significance of data measured by smart meters for network load modeling.

## 5. Conclusions and Perspectives

We have decided to reverse the natural process, which involves starting from the development of aggregated customer curves to obtain an estimate of the secondary substation load curve. By leveraging the richness of Linky meter data, we have created a way to disaggregate the substation curve from this measurement into customer profiles. Developing an estimation methodology based on data measured by smart meters allows us to better estimate a customer's consumption behavior and its changes. This is crucial in the context of integrating new uses of the grid, such as renewable energy sources and electric vehicles. Unlike classical "bottom-up" methodologies that are highly dependent on measurements with a fine temporal resolution, our developed methodology is based on the occurrence time of the maximum daily power. Additionally, our approach involves a disaggregation at a higher grid level, specifically the secondary substation level. This modeling idea is not discussed in the scientific literature, making this model innovative and potentially opening doors to considering how smart meter data can assist in network sizing, integrating new uses, and adapting to changes in consumer behavior in general.

The next stage in the study and improvement of the model involve two factors. The first is an evolution of the segmentation stage by adding climatic and geographical data to the process. The second is the use of energy measured by Linky meters in the disaggregation process. The evolution of the model can be added to the schematic seen in Section 2.3. This is presented in Figure 10.

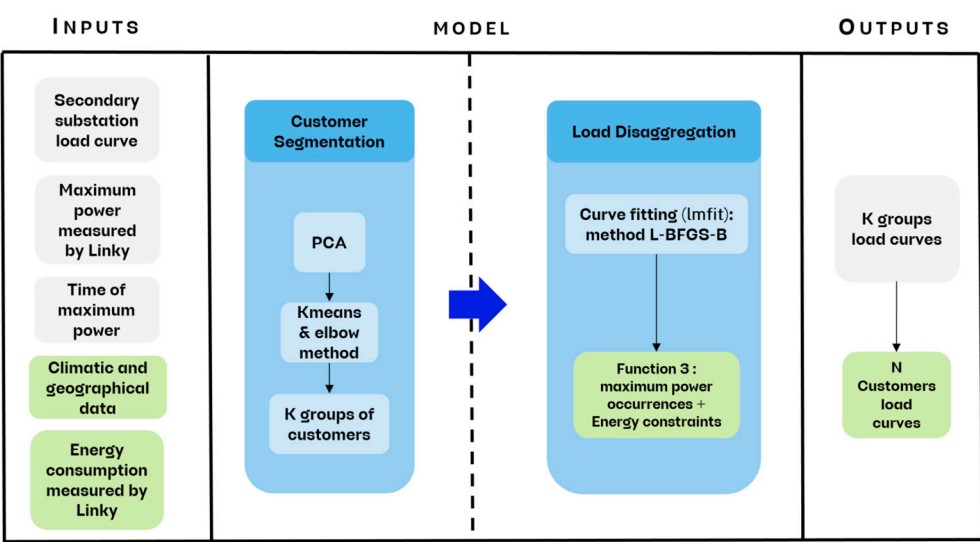

**Figure 10.** Schematic of the disaggregation model with the perspectives.

The addition of this new information to the segmentation step should further enrich the distinction between different types of customers, bringing to light trends that may have been masked previously. This change may be reflected in the disaggregation step using Function (2), as a different number of Gaussians G for each cluster may be found. This could potentially help us better represent the secondary peaks of the load curve of the secondary substation.

Imposing an energy constraint ensures that our result, when viewed from the aggregate of customers, can be distributed to the entire customer base of each group while respecting their consumption. This will further bring the results closer to the reality of consumption in the network. This will allow us to transition from curves aggregated by consumer segments to curves of individual consumers, properly accounting for the energy consumed by them and measured by the Linky meter.

A long-term perspective is the comparison of results produced by the disaggregation methodology with the load curve measurements of low-voltage consumers. However, at the moment, we do not have a secondary substation where all customers have given their consent for the use of this information.

**Author Contributions:** Conceptualization, G.R.M.; methodology, G.R.M.; validation, G.R.M., C.G., M.-C.A.-H. and R.C.; formal analysis, G.R.M.; investigation, G.R.M.; resources, G.R.M.; data curation, G.R.M. and C.G.; writing—original draft preparation, G.R.M.; writing—review and editing, C.G., M.-C.A.-H. and R.C.; visualization, C.G., M.-C.A.-H. and R.C.; supervision, C.G., M.-C.A.-H. and R.C.; project administration, C.G. and R.C.; funding acquisition, R.C. All authors have read and agreed to the published version of the manuscript.

**Funding:** This study is part of a thesis work on the use of Linky smart meter data to improve load knowledge in LV grids. The work is funded by Enedis as part of the SmartGrids Chair of Grenoble INP Fondation and the MIAI Institute: ANRT CIFRE (ANRT 2020/0910) and ANR project 3IA MIAI@Grenoble Alpes (ANR-19-P3IA-0003).

**Data Availability Statement:** The datasets presented in this article are not available because of GDPR. They are associated with a customer panel, who have given their consent to Enedis for the use of their data in the context of Enedis network studies.

**Conflicts of Interest:** Christophe Gay was employed by the company Enedis. The remaining authors declare that the research was conducted in the absence of any commercial or financial relationships that could be construed as a potential conflict of interest.

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
