# Peer review of "Disaggregation Model: A Novel Methodology to Estimate Customers’ Profiles in a Low-Voltage Distribution Grid Equipped with Smart Meters"

_information, doi:10.3390/info15030142_

Round 1

Reviewer 1 Report

Comments and Suggestions for Authors

This manuscript introduces a novel approach to enhance grid hosting capacity by precisely modeling load profiles for low-voltage network consumers amid the imperative energy transition. Traditionally, load curves are derived through distribution system operators' campaigns or appliance curve aggregations. Leveraging smart meters, the authors propose a disaggregation model estimating residential customer load profiles on LV networks. The method utilizes smart meters' maximum power measurements and secondary substation load curves. Evaluated on 48 substations, the model exhibits a 4.91% average symmetric mean average percentage error.

Although this is a critical and timely topic, the existing contributions are not clearly explained to meet the publication requirements and more demonstrations are required. Please consider the following comments in detail:

1-The major concern is whether the proposed research problem addressed in this manuscript is important or realistic. With the increasing prevalence of smart meters, the necessity and significance of employing a disaggregation approach to estimate load profiles come into doubt. The authors should provide a more compelling rationale for why this specific methodology is crucial in the current landscape of expanding smart meter instrumentation. At least for distribution network operators today, the data availability of these data might not be a big issue. For the future, it seems the high-fidelity data are going to be more and more popular.

2-The authors fail to adequately address the impact of external factors on load profiles, which makes the analysis less effective and even not lacking efficient real values. There are quite a lot of potential issues that may take effective and should be discussed at least in the recent work statement. Temperature is always very important to influence the thermal-type loads and influence the load profiles accordingly, and a good example is temperature-sensitive base load in 10.1109/tpwrs.2014.2329485. Some external factors should have decent impacts as well, and a recent evidence is the lockdown restrictions on energy use, you may see 10.1016/j.ijepes.2023.109567. Even technological progress may reshape the load profile, and this statement is validated in 10.1016/j.enbuild.2019.109614. All these factors may not be fully captured in the six categories in the introduction section. To clearly state the knowledge gap, a thorough statement of the relevant works is very important.

3-The user classification methodology presented in this manuscript appears overly simplistic—it seems just a kmeans clustering. The inadequacy lies in the absence of a discerning evaluation of the importance of different indicators, and I believe that this may potential undermine the robustness of the user classification. Will an importance sampling or other approaches be useful in this case and make the segmentation steps more robust? This should be important to load profiles, because these profiles may sometimes dependent on very different groups of influential factors and this make it necessary to include an adaptive factor importance identification.

Why do you only use sMAPE as the metric? There should be some other options and extending this part with other metrics should give a more balanced evidence that the proposed model and algorithm truly work better. Also, why Fig.10 looks so similar as Fig.5?

Comments on the Quality of English Language

Good to read

Author Response

The authors truly thank the reviewer for his/her relevant remarks. A point-by-point response is included in the attached file.

Reviewer 2 Report

Comments and Suggestions for Authors

This paper proposes a disaggregation model for the secondary substation load profile in subsequent customer load profiles, based on a seasonal customer segmentation model and a decomposition model based on a gaussian maximum load identification procedure.

The authors should take into consideration the following issues:

-        Page 4, end of the introductory section: at this point, the authors should clearly declare and describe the original contributions of the paper.

-        Page 4, beginning of section 2: the authors should clearly define the meaning of the “secondary substation” and “customer”. The reviewer suspects that the “secondary substation” refers to a MV/LV transformer, while the “customer” refers to a LV consumer. If so, the authors should add these explanations in the text.

-        Page 5: the authors should clearly identify the segmentation model. Namely, they should present all details concerning the input data (addressed as “maximum power data” in Fig. 2) and how these data are used in the PCA and K-means procedures.

-        Page 6, text after eq. (1): as stated in lines 234-235, one can understand that the secondary substation load curve is the sum of K curves, each one “equal to the substation’s curve”. This is absurd! Rephrase!

-        Page 6, eqs. (1) and (3): the authors mention that “the sum of K curves in eq. 2 is a sum of N Gaussians, as shown in eq. 3”. However, the upper summation limit in 2 is equal to K, while in 3 is equal to N. Use a unique notation!

-        Page 7, sentence “Starting with an analysis for a single customer, all the times at which their maximum power was recorded during the season can be examined”: rephrase or explain this sentence!

-        Page 7, paragraph between Fig. 3 and Fig. 4: what should the reader understand from this statement? That each reading corresponds to the maximum power in 10-minute interval, or each reading is equal to the mean power in the 10-minute interval, as usual, and the profiles in Fig. 4 are usual load profiles? Apparently, as the text in lines 291-297 suggests, Fig. 4 presets normal load profile. Hence, rephrase the paragraph between Figs. 3 and 4!

-        Page 8, eq. (4) and text before it: The authors introduce parameter T with no explanations. Is T a constant for all clusters, or not? May be t+24 hours, or what else?

-        Page 9, Tabel 1: considering the reviewer’s remark for subsection 2.1, page 5, the authors should offer more details concerning the customer segmentation results presented in Table 1. A figure or other representation to show the particular input data for each segment identified by the proposed model would help to better understand the ideas.

Comments on the Quality of English Language

N/A

Author Response

(The authors gave the same response as above.)

Reviewer 3 Report

Comments and Suggestions for Authors

The comments on the paper are included in the attached file.

Author Response

(The authors gave the same response as above.)

Round 2

Reviewer 1 Report

Comments and Suggestions for Authors

I have read the response letter and the revised manuscript. The authors have well addressed my concerns with necessary revisions. Thank you for taking the time and efforts.

Comments on the Quality of English Language

The language quality is generally fine, but a minor text editing might be needed.

Reviewer 2 Report

Comments and Suggestions for Authors

No more comments.

Reviewer 3 Report

Comments and Suggestions for Authors

Thank you for responding to the comments. All the questions I had have been resolved.